# *HOVER*: Hyperbolic Video-text Retrieval

## Abstract

In this paper, we consider video-text retrieval involving action compositions, i.e., a complex video may contain multiple mono-actions such as "sitting up", "opening door", "cooking food", "eating", etc., which is commonly encountered in practice but still with limited research attention. Retrieving such complex videos is challenging and most of the existing methods are observed with substantially degenerated performances. To this end, we propose Hyperbolic Video-tExt Retrieval, HOVER, to model the hierarchical semantic relationships between videos and texts by jointly embedding them into a low-dimensional hyperbolic space. Specifically, a video with action compositions is first decomposed longitudinally into an action tree with mono-action leaf or child nodes and increasingly complex parent nodes. Then, we correspondingly condense the tree-like, chain-like semantic structures and the temporal dependencies between videos into the hyperbolic space based on hyperbolic distance, norm, and relative cosine similarity. Experimental results show that HOVER substantially outperforms its Euclidean counterparts, especially when label size is small in which a performance gain of **28.83%** is achieved. Further, the learned hyperbolic video-text embeddings well generalize to new datasets containing complex videos with varied-level action compositions.

## 1 Introduction

Video-text retrieval is one of the fundamental tasks in computer vision. With the explosive growth of video content on the Internet, video-text retrieval plays an increasingly important role in applications such as search engines, social media, and video websites. The key to video-text retrieval lies in good video/text representation that captures their semantics in a fine-grained manner. Existing methods typically ignore their *is-a* semantic relationship, i.e., a mono-action video/text is semantically affiliated to a complex video/text with multiple actions or semantics. As shown in Fig. 1(a), without capturing such *is-a* relationship, it's difficult for existing methods to distinguish mono-action videos from their compositions in the Euclidean space. Such compositional video-text retrievals are commonly encountered in practice. However, due to the expense leap of annotating such complex videos, existing benchmark datasets typically contain only a small size of compositional annotations as shown in Fig. 1(b), which further leads to substantially degenerated performance in practice.

In this work, we propose Hyperbolic Video-tExt Retrieval (HOVER), which embeds videos/texts into a shared hyperbolic space with their *is-a* semantic relationship explicitly encoded. Instead of embedding each video-text pair separately, HOVER represents the videos and texts to be retrieved in tree-like structures. As shown in Fig. 2, with mono-action videos/texts as the leaf or child nodes, videos/texts with increasing action compositions grow as parent nodes, forming a hierarchical semantic structure. Recent studies Peng et al. (2021); Yang et al. (2022a); Khrulkov et al. (2020b) have demonstrated that the hyperbolic space serves as a continuous analog of discrete trees, making it highly suitable for modeling data that possesses inherently tree-like layouts, such as hierarchical structures or power-law distributed data. The main advantages of hyperbolic space on tree-like data representation lie in a hierarchical geometric prior Krioukov et al. (2010); Nickel & Kiela (2017), low distortion Sarkar (2011); Sala et al. (2018), and a small generalization error bound Suzuki et al. (2021a;b). By jointly embedding the two tree-like structures into a hyperbolic space, HOVER attains fine-grained video/text representation which facilitates distinguishing mono-action videos/texts from their compositions.

Specifically, HOVER aligns videos and texts in a shared hyperbolic space where their semantic *is-a* relationship and temporal dependency are correspondingly represented. i) For *is-a* semantic rela-

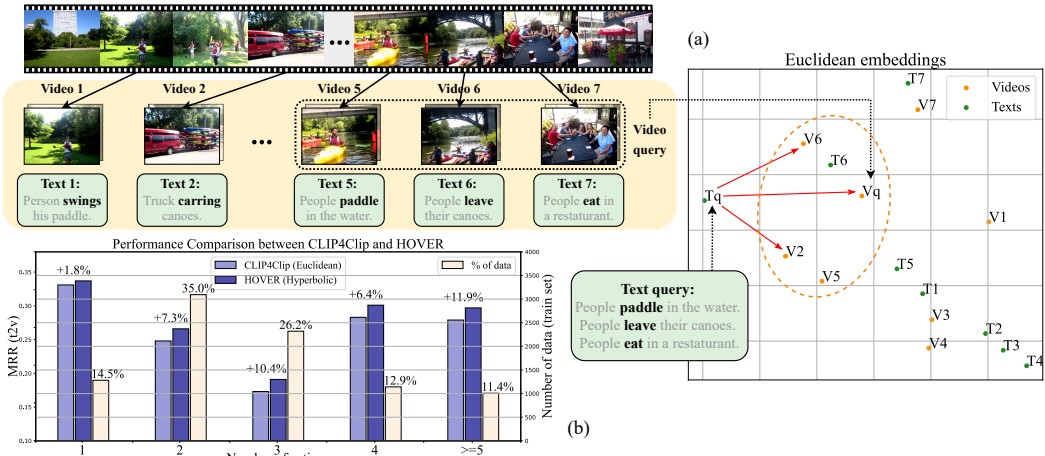

Figure 1: (a) Limitations of existing methods in text-to-video retrieval. Instead of the $V_q$, they tend to prioritize mono-action videos $V_2$ and $V_6$ for the compositional query text string $T_q$. (b) As the composition order grows, the label size shirks in the benchmark dataset ActivityNet.

tionship, two types of structures are considered, a) tree-like structures, where a parent video/text contains two children, and b) chain-like structures, in which a parent video/text contains only one child. . ii) Temporal dependency is essential for action recognition. To capture the temporal relationship between actions, the video/text embedding vectors are further optimized such that the videos/texts observed earlier in a complex video have a hyperbolic angle with smaller cosine values. We summarize our primary contributions as follows:

- We propose to address video-text retrieval involving action compositions by encoding their hierarchical semantic relationship in hyperbolic space.

- We propose to jointly embed videos and texts into Poincaré ball in a cross-modality manner that distinguishes tree-like and chain-like semantic dependencies.

- We propose to encode video-text temporal dependency in Poincaré ball via relative cosine distances.

- Extensively experimenting on benchmark datasets, the numeric results demonstrate the substantial advantages of HOVER over existing methods, especially when the label size is small, and the learned hyperbolic video/text representation is well generalized to datasets with varied action compositions.

## 2 RELATED WORK

### 2.1 VIDEO-TEXT RETRIEVAL

Recent studies on video-text retrieval mainly follow two paradigms of identifying a video-text pair: embed videos and texts with two separate encoders and achieve retrieval via cosine similarity, or learn a joint pairwise similarity via cross-modality architectures. CLIP Radford et al. (2021) provided an essential cross-modal learning paradigm for visual tasks, and has already shown a good generalization on video contents. A series of works like CLIP4Clip Luo et al. (2022), CLIP2Video Fang et al. (2021), X-CLIP Ma et al. (2022), CLIP-ViP Xue et al. (2022) adopt CLIP as a base pretrained model, and transfer image knowledge to video tasks. Other works give additional perspectives besides utilizing image-pretrained models: MDMMT Dzabraev et al. (2021) built a multimodal transformer to fuse video and text features. Frozen Bain et al. (2021) introduced space-time self-attention blocks optimized to temporal video data. X-Pool Gorti et al. (2022) introduced cross-modal attention to reason between a text and each frame of a video. DiffusionRet Jin et al. (2023) made the exploration from generalization, and exerted the diffusion mechanism after modeling the correlation between videos and texts as their joint probability.

Additionally, there are also certain works focusing on building general fundamental models for videos, and consider video-text retrieval as one of the sub-tasks for video understanding. HCMI Jiang et al. (2022) considers the multi-level cross-modal connection between videos and texts on different scales of granularity. They also built a hierarchical semantic structure, but different from our work, they proposed to learn these relations implicitly inside the network and didn't consider the practical semantics of internal tensors throughout the network. InternVideo Wang et al. (2022) adopted self-supervised video learning in a generative and discriminative manner. VALOR Chen et al. (2023) designed pretext tasks to pretrain the model through alignment and captioning. UMT Li et al. (2023) proposed a training-efficient method to learn masked low-semantics video tokens from other unmasked teachers.

## 2.2 HYPERBOLIC REPRESENTATION LEARNING

Hyperbolic spaces have gained significant attention for representing hierarchical data in a wide range of domains Chami et al. (2019); Liu et al. (2019); Yang et al. (2022b; 2021); Nickel & Kiela (2017); Balazevic et al. (2019); Guo et al. (2022); Ghadimi Atigh et al. (2021); Ganea et al. (2018b); Shimizu et al. (2020); Chen et al. (2021)in recent years. Unlike Euclidean spaces, which exhibit polynomial expansion as the radius increases, hyperbolic spaces demonstrate exponential expansion Krioukov et al. (2010). This exponential expansion proves particularly advantageous for hierarchical structures, such as trees, where the number of nodes grows exponentially with increasing depth.

Hyperbolic representation was initially explored in the context of text data Ganea et al. (2018b); Shimizu et al. (2020); Ganea et al. (2018a); Bai et al. (2021); Nickel & Kiela (2017); Balazevic et al. (2019). Textual data naturally exhibit a hierarchical structure, as words can be categorized into more specific subclasses, and sentences can be further broken down into phrases and words. Although recent research has explored the effectiveness of hyperbolic geometry in various computer vision tasks Khrulkov et al. (2020b); Ermolov et al. (2022), they mainly suffer from two weaknesses: 1) The hierarchical structure of the visual contents relies heavily on their textual counterpart. In works on visual tasks like image classification Liu et al. (2020); Ghadimi Atigh et al. (2021), detection Lang et al. (2022), and action recognition Long et al. (2020), the hierarchical structure of texts is first learned, and visual contents are matched to text prototypes. The visual contents themselves do not draw an inherent hierarchical structure. 2) The hierarchical structure in visual contents is learned implicitly. The inherent hierarchical structure of visual contents is not learned locally between samples, but is instead expressed on a global scale with measurements such as hyperbolicity Khrulkov et al. (2020a) and uncertainty estimation Atigh et al. (2022). Motivated by these considerations, our work focuses on explicitly constructing the hierarchical semantic structure and applying it to both video and text samples. We aim to address the limitations of relying on textual information for visual hierarchical structure and the implicit learning of visual content's hierarchical properties.

## 3 PRELIMINARY

Hyperbolic geometry is a type of non-Euclidean geometry characterized by a constant negative curvature, which measures how a geometric object deviates from Euclidean space. Hyperbolic space encompasses various models Peng et al. (2021); Yang et al. (2022a), and these models are isometric to each other. In our study, we employ the Poincaré ball model as the basis for our approach. However, it is important to note that our methodology is not limited to any particular model and can be easily adapted to other models as well.

An $n$-dimensional Poincaré ball model with negative curvature $-c(c > 0)$ is defined as a Riemannian manifold $(\mathbb{D}^n, g^{\mathbb{D}})$ where the manifold $\mathbb{D}^n := \{x \in \mathbb{R}^n : \|x\| < 1/c\}$ is an $n$-dimensional open ball, and $g^{\mathbb{D}}$ is its Riemannian metric:

$$g^{\mathbb{D}} = \lambda_x^2 g^E, \;\; where \;\; \lambda_x = \frac{2}{1 - c\|x\|^2}, \;\; g^E = I_n. \tag{1}$$

We use an exponential map to project vectors in the Euclidean space into the Poincaré Ball. The Euclidean space is seen as the tangent space $\mathcal{T}_x \mathbb{D}^n$ of the manifold $\mathbb{D}^n$ at reference point $x$. As we choose $x = \mathbf{0}$ as the reference point, a tangent vector $v \in \mathcal{T}_x \mathbb{D}^n \backslash \{\mathbf{0}\}$ can be projected into the Poincaré Ball with curvature $c$ by:

$$\exp_{\mathbf{0}}^c(v) = tanh(\sqrt{c}\|v\|)\frac{v}{\sqrt{c}\|v\|}. \tag{2}$$

The distance between vectors $x, y \in \mathbb{D}^n$ in the Poincaré Ball is defined as the length of the geodesic between them, *i.e.* the shortest curve between the two points:

$$d_{\mathbb{D}}(x, y) = \frac{1}{\sqrt{c}}cosh^{-1}(1 + 2\frac{c\|x - y\|^2}{(1 - c\|x\|^2)(1 - c\|y\|^2)}). \tag{3}$$

Since the Poincaré Ball is conformal to the Euclidean space, the cosine distance between two vectors $u, v \in \mathcal{T}_x\mathbb{D}^n\backslash\{\mathbf{0}\}$ is the same as that in the Euclidean space:

$$cos(\angle(u, v)) = \frac{g_{\mathbf{0}}^{\mathbb{D}}(u, v)}{\sqrt{g_{\mathbf{0}}^{\mathbb{D}}(u, u)}\sqrt{g_{\mathbf{0}}^{\mathbb{D}}(v, v)}}. \tag{4}$$

## 4 METHOD

### 4.1 OVERVIEW

Given a set of videos $\mathcal{V} = \{v_i\}_{i=1}^M$ and a set of text descriptions $\mathcal{T} = \{t_i\}_{i=1}^M$, for each video, video-text retrieval aims to find the best-matching text, and vice versa for texts. Typically, a pairwise similarity function $s(v_i, t_j)$ is learned such that the matched text for the query video is prioritized.

HOVER aligns video and text in the shared hyperbolic space, which consists of three components: 1) video-text semantic alignment, which matches the annotated video-text pairs; 2) hierarchical structure embedding, which encodes the cross-modality semantic structure via hyperbolic distances; 3) video/text temporal encoding, which captures the sequential relationship among videos/text.

### 4.2 VIDEO-TEXT SEMANTIC ALIGNMENT

We first match the annotated video-text pairs in the hyperbolic space. As shown in Fig. 2, the video/text representation learning consists of two parts: a) unsupervised pre-training in Euclidean space and b) further fine-tuning with additional hyperbolic transformations. In the Euclidean space, we follow CLIP4Clip Luo et al. (2022) to encode videos and texts with the pre-trained CLIP Radford et al. (2021). For the input text, a deep Transformer encoder network Vaswani et al. (2017) is employed to obtain the text feature. The Euclidean deep features are then projected into the Poincaré Ball via exponential mapping and are followed by hyperbolic transformations.

The alignment of annotated video-text pairs is attained via a standard training objective. Specifically, with the labeled video-text pairs as positives and unreported as negatives, the alignment objective $\mathcal{L}_{align}$ is:

$$\begin{aligned}
\mathcal{L}_{align}^{v2t} &= -\frac{1}{B}\sum_i^B log\frac{exp(-d_{\mathbb{D}}(v_i^{\mathbf{H}}, t_i^{\mathbf{H}}))}{\sum_{j=1}^B exp(-d_{\mathbb{D}}(v_i^{\mathbf{H}}, t_j^{\mathbf{H}}))}, \\
\mathcal{L}_{align}^{t2v} &= -\frac{1}{B}\sum_i^B log\frac{exp(-d_{\mathbb{D}}(v_i^{\mathbf{H}}, t_i^{\mathbf{H}}))}{\sum_{j=1}^B exp(-d_{\mathbb{D}}(v_j^{\mathbf{H}}, t_i^{\mathbf{H}}))},
\end{aligned} \tag{5}$$

where $v_i^{\mathbf{H}}$ and $t_i^{\mathbf{H}}$ correspondingly denote the hyperbolic embeddings of the $i$-th video and the $j$-th text in a training batch of size $B$, and $d_{\mathbb{D}}(v^{\mathbf{H}}, t^{\mathbf{H}})$ is the hyperbolic distance in the Poincaré Ball.

### 4.3 HIERARCHICAL STRUCTURE EMBEDDING

In this section, we embed the hierarchical relationship between videos and texts in hyperbolic space. Two types of semantic structures are considered: semantic trees and semantic chains.

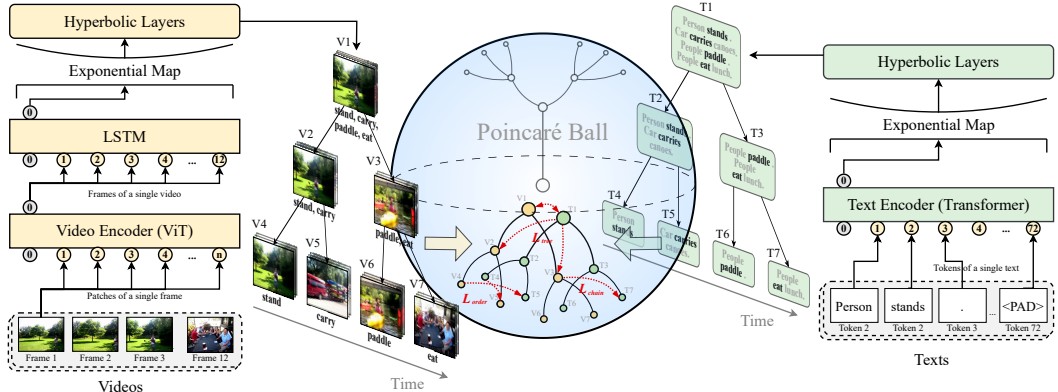

Figure 2: Architecture of HOVER. It first decomposes complex videos and the corresponding texts longitudinally to derive the semantic structures that are jointly embedded into the Poincaré Ball with both hierarchical relationship and temporal dependency encoded.

### 4.3.1 SEMANTIC TREE

Semantic trees are frequently encountered in the analysis of videos and texts, where a parent video or text may contain multiple child videos or texts. We aim to embed the semantic tree in a cross-modality manner by considering the video-text "is-a" relationship. We observe this approach with significant improvements compared to embedding text and video trees separately.

To construct the embedding, we assign positive labels to videos and texts that have an "is-a" relationship (either as parent or child) and negative labels otherwise. The objective function $\mathcal{L}_{tree}$ for embedding the semantic tree can be defined as follows:

$$
\begin{aligned}
\mathcal{L}_{tree}^{v2t} &= - \sum_{(p,q)\in E} log \; \frac{exp(-d_\mathbb{D}(v_p^{\mathbf{H}}, t_q^{\mathbf{H}}))}{\sum_{q'\in N(p)} exp(-d_\mathbb{D}(v_p^{\mathbf{H}}, t_{q'}^{\mathbf{H}}))}, \\
\mathcal{L}_{tree}^{t2v} &= - \sum_{(p,q)\in E} log \; \frac{exp(-d_\mathbb{D}(t_p^{\mathbf{H}}, v_q^{\mathbf{H}}))}{\sum_{q'\in N(p)} exp(-d_\mathbb{D}(t_p^{\mathbf{H}}, v_{q'}^{\mathbf{H}}))},
\end{aligned}
\tag{6}
$$

where $v^{\mathbf{H}}$ and $t^{\mathbf{H}}$ denote the hyperbolic embedding vectors for videos and texts, respectively. $E$ represents the set of video/text pairs with the "is-a" relationship, and $N(p)$ denotes a randomly selected set of negative videos or texts. It is important to note that in the hyperbolic space, positioning the parent (or root) node of a tree-like data structure at the hyperbolic origin results in a relatively short distance between the parent node and all other nodes. This is due to the parent node's norm being zero. Conversely, child (or leaf) nodes tend to be positioned closer to the outer edges of the hyperbolic ball. This is because as the norm approaches one, the distance between points rapidly increases. Hence, the objective function $\mathcal{L}_{tree}$ guides the model to embed parent videos or texts with smaller norms compared to their children, which aligns with findings from prior research Nickel & Kiela (2017); Sala et al. (2018).

### 4.3.2 SEMANTIC CHAIN

In practical scenarios, it is common to encounter action-compositional video/text instances where the parent entity is associated with only one child mono-action video/text, resulting in a directed semantic chain. To incorporate this information into the embedding process, we explicitly encourage the parent video/text to have smaller norms compared to its children through the utilization of the objective function $\mathcal{L}_{chain}$:

$$\mathcal{L}_{chain}^{v2t} = -\alpha_1 \max\left(\left\|v_c^{\mathbf{H}}\right\| - \left\|t_p^{\mathbf{H}}\right\| + \epsilon_1, 0\right),$$
$$\mathcal{L}_{chain}^{t2v} = -\alpha_1 \max\left(\left\|t_c^{\mathbf{H}}\right\| - \left\|v_p^{\mathbf{H}}\right\| + \epsilon_1, 0\right),$$

(7)

where $c$ and $p$ correspondingly denote child video/text, $\epsilon_1$ is a threshold that controls the parent-child norm gap. Besides, we introduce the coefficient $\alpha_1$ to control the impact of the norm difference between parent and child embeddings as previous work Nickel & Kiela (2018).

## 4.4 VIDEO/TEXT TEMPORAL ENCODING

Capturing temporal dependencies is crucial for comprehending videos, both in terms of individual actions and their compositions. To address this, we extend our approach to embed the temporal dependency between videos in the hyperbolic space. Specifically, we encourage videos/texts that occur earlier to have smaller angles in the hyperbolic embedding space. This optimization is achieved through the objective function $\mathcal{L}_{order}$:

$$\mathcal{L}_{order}^{v2t} = -\alpha_2 \max(cos(\angle(v_{c_l}^{\mathbf{H}}, \vec{\mathbf{j}})) - cos(\angle(t_{c_r}^{\mathbf{H}}, \vec{\mathbf{j}})) + \epsilon_2, 0),$$
$$\mathcal{L}_{order}^{t2v} = -\alpha_2 \max(cos(\angle(t_{c_l}^{\mathbf{H}}, \vec{\mathbf{j}})) - cos(\angle(v_{c_r}^{\mathbf{H}}, \vec{\mathbf{j}})) + \epsilon_2, 0),$$

(8)

where $c_l$ denotes a child video/text observed prior to child video/text $c_r$ in the complex video with action compositions. $c_l$ represents a child video/text observed prior to the child video/text $c_r$ within a complex video containing action compositions. $\vec{\mathbf{j}}$ is a learnable reference base. The cosine function is utilized to measure the angular similarity between the hyperbolic embedding vectors. The parameter $\epsilon_2$ serves as a threshold controlling the desired difference in angles between child video/text pairs. The coefficient $\alpha_2$ is introduced to regulate the influence of the angle difference during training.

Combining the above objectives, we obtain the final loss $\mathcal{L}_{joint}$:

$$\mathcal{L}_{joint} = \frac{1}{2}(\mathcal{L}_{align}^{v2t} + \mathcal{L}_{align}^{t2v}) + \frac{\beta}{2}(\mathcal{L}_{tree}^{v2t} + \mathcal{L}_{tree}^{t2v} + \mathcal{L}_{chain}^{v2t} + \mathcal{L}_{chain}^{t2v} + \mathcal{L}_{order}^{v2t} + \mathcal{L}_{order}^{t2v}), \quad (9)$$

where $\beta$ denote the hyper-parameters that control the contribution of hierarchical semantic embedding.

## 5 EXPERIMENTS

### 5.1 DATASETS

Based on video moment retrieval datasets, we construct hierarchical video/text relationships. In an original video moment retrieval dataset, each video is annotated with a series of text descriptions, along with their timestamps. These video clips in a single video serve as the leaf nodes of the tree structure, and we build the complete hierarchical structure on top of them, as is shown in Fig. 3.

We built the hierarchical datasets based on two datasets: **ActivityNet-1.3** Caba Heilbron et al. (2015), which consists of 20K YouTube videos with 100K annotated moments; **Charades** Gao et al. (2017), which is composed of 6672 videos of indoor activities annotated with 11767 moments. The ActivityNet-based datasets are used for training and evaluation, and the Charades-based dataset is only used for evaluation due to its relatively simple tree structures. More details are provided in the Supplementary Materials.

### 5.2 PERFORMANCE METRICS

We evaluate the retrieval performance in both directions: text-to-video retrieval (t2v) and video-to-text retrieval (v2t), with the following metrics:

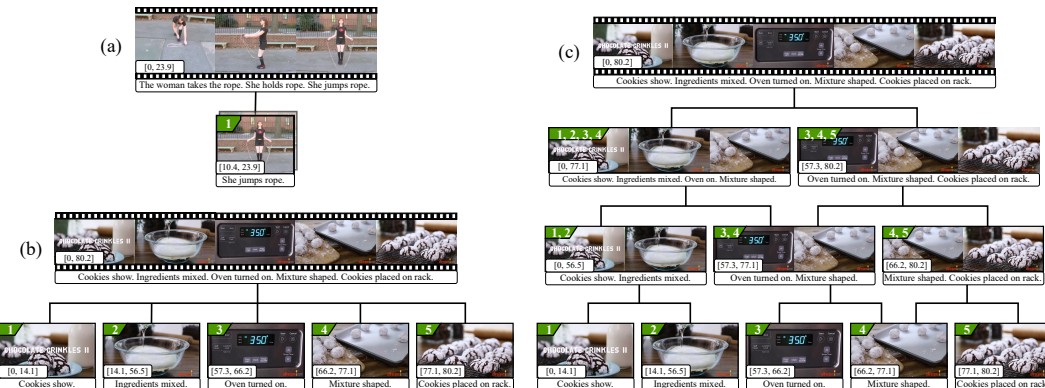

Figure 3: Three types of hierarchical relationships are considered in the experiments: (a) Single-Leaf Trees: a video chain with one root node and one leaf node. (b) Multi-Leaf Trees, Shallow: video trees with a height of 2. All the video clips are treated as leaf nodes and the final one aggregating all leaf nodes acts as the root. (c) Multi-Leaf Trees, Deep: video trees with all the video clips as leaf nodes, and adjacent leaf nodes are concatenated to form their corresponding parent node.

**Recall** ($\uparrow$)    Recall at rank $K$ (R@$K$) represents the percentage of correct results appearing in the top K responses for all video or text queries. We report the results for $K \in \{1, 5, 10\}$.

**MRR** ($\uparrow$)    Mean reciprocal rank (MRR) is a relatively smoother metric than recall. It measures the average of the reciprocal ranks of correct results for all queries $Q$, $MRR = \frac{1}{|Q|} \sum_{i=1}^{|Q|} \frac{1}{rank_i}$.

## 5.3 IMPLEMENTATION DETAILS

For the Euclidean setting of the model, we follow the same implementations as CLIP4Clip Luo et al. (2022). For the hyperbolic part, we set the learning rate at 1e-4, with the same optimizer as the Euclidean setting. All hyperbolic linear layers are initialized with identity matrices for weights and zero for biases. The additional linear layers in the Euclidean baseline are implemented with the same setting as the hyperbolic linear layers. We use a negative curvature with $c = 1$ for the exponential map and hyperbolic layers. The max length for the text encoder is set as 72, and the max frame number for the video encoder is set as 12. We set $\alpha_1 = 1, \alpha2 = 0.2$ , $\epsilon_1 = \epsilon_2 = 0.1, \beta = 1$, respectively. All experiments are conducted on 2 NVIDIA Tesla V100 GPUs with 32GB of memory.

## 5.4 COMPARISON TO STATE-OF-THE-ART METHODS

We first compare our hyperbolic method with Euclidean SoTA directly on ActivityNet (Hierarchy). Since the performance of hyperbolic representation is also influenced by the corresponding Euclidean baseline, we choose two models: Frozen Bain et al. (2021) and CLIP4Clip Luo et al. (2022) (The LSTM type with the best performance in their work) that are trained on video-text retrieval datasets. In the few-shot setting, we train our model with only 10% of the training data and then test on the testing set. As reported in Tab. 1, the hyperbolic method outperforms all the Euclidean SoTA. Compared to the Euclidean method with the best performance, there is a 16.6% performance improvement in the full setting and +28.8% in the few-shot setting. Additionally, our method also works better in video-to-text scenarios, where Euclidean methods achieve weaker performance than text-to-video retrieval.

We also compare the performance of our method and SoTA methods in videos with different composition orders of actions, as shown in Tab. 1. The second column "leaf" represents the number of leaf nodes in a single video tree. Overall, the best performance is achieved in high-order trees containing 3 leaf nodes, with +28.83% in MRR for text-to-video retrieval over the Euclidean baseline.

Note that the performance drop on single leaf data is due to the lack of compositional videos for training. When trained on the entire dataset, HOVER achieves +1.81% MRR compared to CLIP4Clip, and +7.26%, +10.40%, +6.36%, +6.45% on trees with 2, 3, 4, ≥5 leaf nodes respectively.

Table 1: Comparison results of the proposed HOVER and SoTA methods (CLIP4Clip Luo et al. (2022), Frozen Bain et al. (2021)) on ActivityNet (Hierarchy).

| Dataset | Leaf | Method | t2v R@1 | R@5 | R@10 | v2t R@1 | R@5 | R@10 | MRR(t2v) | ΔMRR |
|---------|------|--------|---------|-----|------|---------|-----|------|----------|------|
| Full | 1 | CLIP4Clip | **8.6** | **22.8** | **31.8** | 5.9 | 19.3 | 28.6 | **0.164** | -9.76% |
| | | HOVER | 7.6 | 19.4 | 28.4 | **8.6** | **22.8** | **32.2** | 0.148 | |
| | 2 | CLIP4Clip | 6.5 | 20.1 | 28.9 | 5.9 | 19.5 | 28.7 | 0.141 | +9.22% |
| | | HOVER | **7.5** | **21.7** | **30.2** | **8.1** | **23.5** | **33.2** | **0.154** | |
| | 3 | CLIP4Clip | 5.7 | 21.0 | 31.0 | 4.8 | 18.3 | 28.4 | 0.138 | +12.32% |
| | | HOVER | **6.8** | **23.6** | **34.1** | **7.0** | **23.9** | **33.7** | **0.155** | |
| | 4 | CLIP4Clip | 7.9 | 23.7 | 33.2 | 7.0 | 22.1 | 32.1 | 0.163 | +15.95% |
| | | HOVER | **9.1** | **29.3** | **38.9** | **10.3** | **30.1** | **40.1** | **0.189** | |
| | ≥5 | CLIP4Clip | 8.5 | 26.0 | 39.1 | 7.4 | 22.5 | 33.0 | 0.180 | +8.33% |
| | | HOVER | **9.2** | **28.8** | **42.9** | **11.5** | **31.9** | **44.2** | **0.195** | |
| | Overall | Frozen | 4.8 | 16.8 | 25.3 | 4.8 | 15.8 | 23.9 | 0.117 | +16.56% |
| | | CLIP4Clip | 6.9 | 22.1 | 32.2 | 5.9 | 19.8 | 19.6 | 0.151 | |
| | | HOVER | **7.7** | **24.2** | **34.6** | **8.6** | **25.7** | **35.9** | **0.176** | |
| Few-shot | Overall | Frozen | 3.6 | 11.7 | 17.8 | 2.7 | 9.2 | 14.3 | 0.084 | +28.83 % |
| | | CLIP4Clip | 4.7 | 15.9 | 23.8 | 3.9 | 14.2 | 21.9 | 0.111 | |
| | | HOVER | **6.5** | **21.0** | **30.6** | **7.3** | **21.6** | **30.5** | **0.143** | |

## 5.5 Zero-shot Transfer to Novel Datasets

We evaluate the generalization of the learned video/text representation to novel datasets, *i.e.*, Charades (Hierarchy), MSR-VTT Xu et al. (2016) and MSVD Chen & Dolan (2011). Charades (Hierarchy) is built based on the original Charades dataset, which is similar to ActivityNet (Hierarchy). MSR-VTT and MSVD are widely-used datasets in video-text retrieval tasks. As shown in Tab. 2, HOVER outperforms the best Euclidean method by +10.3%, +26.4%, and +12.4% in MRR for the text-to-video retrieval on Charades (Hierarchy), MSR-VTT and MSVD dataset, respectively.

Table 2: Results of zero-shot transfer of HOVER and Euclidean SoTA methods (CLIP4Clip Luo et al. (2022), Frozen Bain et al. (2021)) from ActivityNet (Hierarchy) to Charades (Hierarchy), MSR-VTT, and LSMDC. Datasets marked with † are widely-used for video-text retrieval.

| Dataset | Method | t2v R@1 | R@5 | R@10 | v2t R@1 | R@5 | R@10 | MRR(t2v) | ΔMRR |
|---------|--------|---------|-----|------|---------|-----|------|----------|------|
| Charades (Hierarchy) | Frozen | 0.6 | 3.7 | 6.0 | **1.1** | 3.5 | 5.6 | 0.029 | +10.34% |
| | CLIP4Clip | 0.7 | 2.8 | 5.0 | 0.8 | 2.6 | 4.5 | 0.025 | |
| | HOVER | **1.0** | **4.7** | **8.1** | 1.0 | **3.9** | **6.3** | **0.032** | |
| MSR-VTT† | Frozen | 21.8 | 42.3 | 54.2 | 21.4 | 44.5 | 55.0 | 0.322 | +26.40% |
| | CLIP4Clip | 11.9 | 24.5 | 32.0 | 13.5 | 31.9 | 41.8 | 0.188 | |
| | HOVER | **30.4** | **51.2** | **60.7** | **25.1** | **48.5** | **59.2** | **0.407** | |
| MSVD† | CLIP4Clip | 35.0 | 64.8 | 75.1 | 42.5 | 73.3 | 82.2 | 0.484 | +12.40% |
| | HOVER | **41.3** | **69.9** | **79.8** | **59.6** | **82.8** | **86.8** | **0.544** | |

## 5.6 Cross-order Generalization

Cross-order generalization is commonly encountered in video-text practice. While these complex videos are prevalent in real-world samples, only simplified queries in video-text retrieval datasets exist. Hence, to evaluate the generalization ability of hyperbolic representation, we make a comparison between our method and the Euclidean baselines by training solely on 2-order video trees and evaluating performance on high-order compositions. The second-order tree consists of only one root node and two leaf nodes. When training the model on 2-order trees, we only provide videos with less obvious hierarchical structures as training data. This scenario reflects the practical application where models are trained on datasets primarily composed of low-order compositions, yet evaluated on real-world videos with high-order compositions. As shown in Tab. 3, our method shows significant improvements in all orders compared with the Euclidean baselines. Notably, the best performance is achieved on trees containing 4 leaf nodes with +41.1% in MRR over the Euclidean

baselines for text-to-video retrieval, indicating the robustness of our method in handling high-order action compositions.

Table 3: Results of cross-order generalization on ActivityNet (Hierarchy), compared with CLIP4Clip Luo et al. (2022). The models are trained using only 2-order trees but evaluated across all orders.

| Leaf | Method | t2v R@1 | R@5 | R@10 | v2t R@1 | R@5 | R@10 | MRR (t2v) | ΔMRR |
|------|--------|---------|-----|------|---------|-----|------|-----------|------|
| 1 | CLIP4Clip | 8.0 | 20.7 | 29.4 | 6.3 | 18.2 | 27.0 | 0.152 | -5.92% |
| | HOVER | **7.2** | **18.7** | **27.1** | **8.7** | **20.6** | **29.8** | **0.143** | |
| 2 | CLIP4Clip | 5.3 | 16.8 | 25.2 | 5.1 | 17.6 | 26.0 | 0.121 | +19.01% |
| | HOVER | **6.7** | **20.4** | **19.4** | **6.8** | **20.7** | **29.8** | **0.144** | |
| 3 | CLIP4Clip | 5.0 | 19.3 | 28.5 | 4.3 | 17.0 | 26.2 | 0.126 | +11.90% |
| | HOVER | **5.9** | **21.5** | **32.1** | **6.5** | **23.0** | **32.4** | **0.141** | |
| 4 | CLIP4Clip | 5.8 | 18.4 | 28.1 | 5.4 | 18.0 | 26.4 | 0.129 | **+41.09%** |
| | HOVER | **9.1** | **27.2** | **37.8** | **9.5** | **28.5** | **37.6** | **0.182** | |
| ≥5 | CLIP4Clip | 5.8 | 21.5 | 32.2 | 5.2 | 18.4 | 28.7 | 0.142 | +26.76% |
| | HOVER | **8.2** | **26.7** | **39.0** | **11.2** | **31.9** | **42.9** | **0.180** | |

## 5.7 ABLATION STUDY

We investigate the contributions of each proposed component in aligning videos and texts in the hyperbolic space. Apart from the video/text semantic alignment, individual and combined contributions are assessed on all other components: semantic tree embedding, semantic chain embedding, and temporal encoding. As shown in Tab. 4, each proposed component helps boost the performance. Beyond the numeric results, we provide embedding visualizations to further validate their contributions in the Supplementary Materials.

Table 4: Ablation study of the components of HOVER (Align: video-text semantic alignment; Tree: semantic tree embedding; Chain: semantic chain embedding; Time: temporal encoding).

| Align | Tree | Chain | Time | t2v R@1 | R@5 | R@10 | MRR (t2v) |
|-------|------|-------|------|---------|-----|------|-----------|
| ✓ | ✓ | | | 7.6 | 24.0 | 34.5 | 0.163 |
| ✓ | | ✓ | | 7.3 | 23.4 | 33.5 | 0.158 |
| ✓ | ✓ | ✓ | | 7.6 | **24.1** | 34.4 | 0.164 |
| ✓ | ✓ | ✓ | ✓ | **7.8** | 24.0 | **34.5** | **0.164** |

## 6 CONCLUSION

In this work, we propose to learn hierarchical representation for video-text retrieval in hyperbolic space. Due to the limitations of existing Euclidean methods, which overlook the multi-level semantic structure inherent in complex videos, we propose to embed videos and texts into a shared hyperbolic space, where the hierarchical semantic structures are explicitly encoded. To align videos and texts in the hyperbolic space, three major components are proposed: 1) video-text semantic alignment; 2) hierarchical structure embedding; 3) video/text temporal encoding. Experimental results validate the superiority of our method over the Euclidean counterparts and the learned video/text representation is shown with strong generalization ability to complex videos with high-order action compositions.

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
