# OpenReview forum: "HOVER: Hyperbolic Video-text Retrieval"
_ICLR.cc/2024/Conference — ICLR 2024 Conference Withdrawn Submission_

### Official Review · Reviewer_BhHY · 2023-10-29

**Soundness:** 2 fair
**Presentation:** 2 fair
**Contribution:** 2 fair
**Rating:** 3
**Confidence:** 5

**Summary:**

The authors propose Hyperbolic Video-tExt Retrieval to model the Is-A semantic relationships between videos and texts.
Specifically, a video with action compositions is first decomposed longitudinally into an action tree with mono-action leaf or child nodes and increasingly complex parent nodes. Then, the is-a semantic relationship in videos/texts is represented in the hyperbolic space by employing hyperbolic norm constraints.

**Strengths:**

- A new method for compositional video retrieval.

- Show somewhat better results than previous methods.

**Weaknesses:**

- What's the difference between Is-A semantic relationship and partially relevant?
The authors are recommended to make necessary explanations on this issue.
Most importantly, the authors should make comprehensive and fair comparisons between partially relevant retrieval[A][B] and multi-event retrieval[C].
Additionally, the HOVER could be performed after the hierarchical datasets were built while partially relevant retrieval did not need the operation.

- No comprehensive experiments on the sensitivity of the proposed parameters.

- Reported as Table 2, larger improvement on non-hierarchical datasets: MSR-VTT and MSVD needs necessary analysis.

- What is the difference between the chain-like struture and tree-like structure in the design of losses?
The authors are recommended to make necessary explanations on the non-improvement of the performance when the chain-loss is added, presented as Table 4.
The ablation with only Align loss is also recommended to present here for further understanding this work.

- Other minor issues:

     -- No LSMDC in Table 2.

     -- Double quotation marks for is-a are incorrect.

    -- Deleting the extra full stop below the caption of Figure 2.



references:

A. Partially Relevant Video Retrieval, ACM MM 2022

B. Progressive Event Alignment Network for Partially Relevant Video Retrieval, ICME 2023

C. Multi-Event_Video-Text_Retrieval, ICCV 2023.

**Questions:**

Presented as weaknesses.

---

### Official Review · Reviewer_AjjV · 2023-10-30

**Soundness:** 3 good
**Presentation:** 3 good
**Contribution:** 3 good
**Rating:** 5
**Confidence:** 4

**Summary:**

This paper uses hyperbolic space to represent the Is-A relationship in compositional actions, improving video comprehension. Apart from aligning text and video, the suggested method achieves: 1) Using the Poincaré ball, study the representation of video and text while considering their hierarchical relationship. 2) Obtaining information about the ordering of video-text pairs. The suggested model improves across multiple tasks by adequately representing the properties of composing activities.

**Strengths:**

1. The paper is well-written and easy to follow, and the visualizations are nicely designed.
2. The proposed ``semantic-chain'' is novel and nicely handles one common ill-conditioned case in hierarchical structure.
3. Intuitively, the combination of semantic tree, semantic chain, and temporal ordering can nicely capture the structure inherent in compositional actions.
4. Experimental results across a diverse range of tasks provide sound evidence supporting the efficacy of the proposed model.

**Weaknesses:**

1. The novelty is not clearly stated, as many components (text-visual training, hyperbolic embedding, temporal encoding) have been proposed by other models. The authors should highlight their contribution.
2. The absence of the base case(s) in Table 4, for example, when only alignment is utilized and only temporal encoding is deployed.
3. Tables with good numbers are good, but it is less interesting than the analysis and explanation of WHY the proposed model works.

**Questions:**

Please respond to the weaknesses.

---

### Official Review · Reviewer_MxoF · 2023-10-31

**Soundness:** 3 good
**Presentation:** 2 fair
**Contribution:** 2 fair
**Rating:** 3
**Confidence:** 4

**Summary:**

This paper focuses on the challenging task of retrieving complex videos with multiple actions. They introduce HOVER, a method that uses hyperbolic space to model the hierarchical relationships between videos and texts. HOVER outperforms traditional methods, especially when the dataset is small, with a substantial performance gain of 28.83%. It also generalizes well to new datasets with complex videos.

**Strengths:**

1. The proposed method with novel semantic tree, semantic chain, and temporal encoding is quite interesting.
2. The proposed method has better performance.

**Weaknesses:**

1. The writing is poor and the paper is not self-contained. To be honest, this paper investigates a novel task instead of the traditional video-text (paragraph) retrieval and video moment retrieval. So first thing the authors should do is actually present the definition of ‘hierarchical video retrieval’ before introducing any other technical details.
2. It would be interesting to see whether the use of other balls leads to similar or better performance with the proposed novel semantic tree, semantic chain, and temporal encoding.
3. The details of the proposed method are missing. For example, the details of hyperbolic layers, exponential map, and how to calculate the similarity after obtaining the tree.

This paper needs another round of proofreading before submitting to any prestigious conference.

**Questions:**

See weakness

---

### Official Review · Reviewer_P4PE · 2023-10-31

**Soundness:** 3 good
**Presentation:** 3 good
**Contribution:** 3 good
**Rating:** 6
**Confidence:** 4

**Summary:**

This paper introduces a hyperbolic feature space to encode the visual and text embedding for the task of video-text retrieval. Given a complex video sequence consisting of multiple mono-actions, this work embeds videos and texts into Poincare ball with the manner of semantic tree and chain. Compared to the baseline method, this work achieves promising results.

**Strengths:**

1. The entire paper is well-written, the idea is easy to follow and the motivation is well justified.

2. The idea of well-structuring the visual and linguistic embedding into the hyperbolic manifold is novel and interesting, which constructs discriminative feature representations for the complex videos.


3. This work also provides many insights of modeling the long-form sequences in the vision and language feature learning. The temporal dependency in the Poincare ball could potentially improve the longer temporal reasoning in the video understanding.


4. This work has done extensive comparison with the baseline method (clip4clip), which clearly demonstrates the effectiveness of each proposed component in the algorithm. Moreover, compared to the clip4clip, the performance of this work looks very promising.


5. Code is provided which is a plus for the re-implementation.

**Weaknesses:**

1. In the experiment section, this work only compares with the clip4clip. It will be good to compare with other SOTA methods showing the superior performance of this work.

2. The clip4clip is the video-extension model of clip model. When there are multiple leaves, the clip4clip may not have effective mechanism to model longer sequence/text. However, the proposed HOVER formulates a tree/chain to address this. Is there a better way to verify this hypothesis?

3. In some of previous Riemannian manifold works [1, 2], people may face the non-convex issue when doing the optimization. For the stochastic-based optimization used in this work, how to constraint the learned feature always maintains a hyperbolic manifold?


[1] Generalized rank pooling for activity recognition, CVPR17

[2] Discriminative Video Representation Learning Using Support Vector Classifiers, TPAMI

**Questions:**

Mentioned in the weakness